# Morpho-Functional Macular Assessment in a Case of Facioscapulohumeral Muscular Dystrophy: Photoreceptor Degeneration as Possible Cause for Reduced Visual Acuity over Three Years of Follow-Up

**DOI:** 10.3390/diagnostics12122977

**Published:** 2022-11-28

**Authors:** Mariacristina Parravano, Eliana Costanzo, Lucilla Barbano, Pasquale Viggiano, Daniele De Geronimo, Giulio Antonelli, Vincenzo Parisi, Monica Varano, Lucia Ziccardi

**Affiliations:** 1IRCCS—Fondazione Bietti, 00198 Rome, Italy; 2Department of Basic Medical Science, Neuroscience and Sense Organs, University of Bari “Aldo Moro”, 70121 Bari, Italy

**Keywords:** FSHD, macular abnormalities, photoreceptor degeneration

## Abstract

Background: Autosomal-dominant facioscapulohumeral muscular dystrophy (FSHD) is a muscular dystrophy with associated retinal abnormalities such as retinal vessel tortuosity, focal retinal pigment epithelium defect and large telangiectasia vessels. Methods: Case report of an FSHD 16-year-old female referred for blurred vision in both eyes (20/40), evening fever and shoulder muscle weakness over the past month preceding assessment. A multimodal assessment including visual acuity (VA), microperimetry (MP), multifocal electroretinogram (mfERG), optical coherence tomography (OCT), fluorescein angiography (FA) and fundus autofluorescence (FAF) was performed. Results: OCT showed pseudocyst macular abnormalities and disruption of the photoreceptor layer with no signs of macular ischemia/exudation. Macular function showed foveal impairment recorded by mfERG and MP as a reduction of the response amplitude density and retinal sensitivity, respectively. No medical treatment was prescribed. After three years, patient’s VA slightly improved to 20/32. OCT showed resolution of bilateral pseudocyst macular changes and persistence of photoreceptor disruption. By contrast, mfERG recordings remained abnormal for impaired foveal function and microperimetry mean sensitivity was reduced as well. Conclusions: This multimodal assessment showed persistent VA impairment at three years follow-up associated to abnormal foveal function and reduced retinal sensitivity, with spontaneous resolution of morphological macular changes, suggesting a retinal neurodegenerative process on the basis of the disease.

**Figure 1 diagnostics-12-02977-f001:**
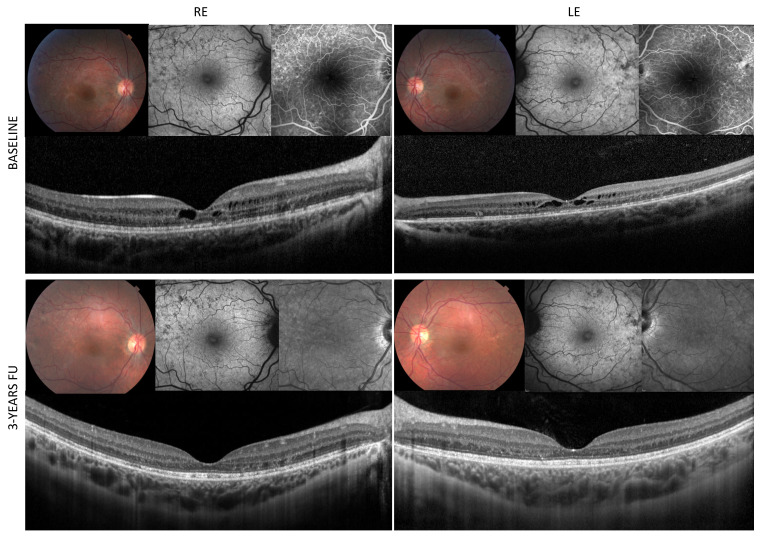
Multimodal imaging assessment of a 16-year-old female affected by Autosomal-dominant facioscapulohumeral muscular dystrophy (FSHD) [1], harboring the molecular variant on chromosome 4q35 (D4Z4 repeat contractions by DNA testing from peripheral blood), referred to the IRCCS-Fondazione Bietti Medical Retina Unit, for blurred vision in both eyes, evening fever and shoulder muscle weakness over the past month preceding assessment. Her visual acuity was 20/40 (Snellen Charts) in both eyes, no ptosis, intraocular pressure and anterior segment were within normal limits, no signs of inflammation in the anterior or posterior chambers was noted. Fundoscopy revealed marked retinal arteriolar tortuosity, RPE defects and foveal thickening in both eyes (1 upper boxes). Macular Spectral-Domain Optical Coherence Tomography (Spectralis HRA+OCT, Heidelberg Engineering, Heidelberg, Germany) scans showed bilateral pseudocyst abnormalities and disruption of the photoreceptor layer (second line boxes). Fundus autofluorescence (FAF) and fluorescein angiography (FA) (Spectralis HRA+OCT, Heidelberg Engineering, Heidelberg, Germany) exhibited widespread RPE defects located in the posterior pole and in the mid periphery with early macular hyperfluorescence without leakage in the late frame and no signs of peripheral ischemia/exudation (, upper boxes). In order to exclude other causes that could have triggered the visual acuity impairment, the patient also performed a visual field exam (30-2) and visual evoked potentials with results falling within normal limits. No medical treatment was prescribed for the observed macular abnormalities at baseline.

After three years, the patient’s visual acuity slightly improved to 20/32 in both eyes. The repeated SD-OCT scans showed spontaneous resolution of bilateral pseudocyst macular changes and persistence of photoreceptor disruption (Figure 1 lower boxes). No peripheral ischemia or exudation developed over time at FA.

Intravitreal treatment with anti-VEGF or laser photocoagulation was not indicated for the pseudocyst macular changes, considering the already reported case of resolution for dynamic spatiotemporal changes of the vascular alterations in FSHD [2]. Moreover, this case report alerts retinal specialists to be cautious and promptly plan treatment for macula edematous appearance in FSHD, since self-resolution over time may happen, and because reduced visual acuity could be secondary to a process independent from retinal oedema or ischemia.

After three years, despite the spontaneous resolution of morphological abnormalities, mfERG recordings (Figure 2B) remained abnormal for impaired foveal function (R1), likely due to photoreceptor degeneration, and microperimetry mean sensitivity was reduced to 15.2 dB in RE and 17.4 in LE as well (Figure 2D).

As a novel finding, in our FSHD patient we encountered reduced visual acuity, likely ascribed to macular abnormalities, as signs of outer retina degeneration. In our case, based on the spontaneous resolution of bilateral macular pseudocyst aspect, not associated to FA signs of vascular exudation or ischemia, as also previously reported [3], we may ascribe the visual acuity impairment to photoreceptor layer degeneration.

Only one previous report [4] investigated on retinal function in two FSHD patients by flash and flicker ERG recordings, demonstrated decreased cone amplitude.

In the present case, for the first time, we combined psychophysical (visual acuity), morphological (SD-OCT, FAF, FA) and functional (mfERG and microperimetry) assessment for studying the macular area.

This multimodal assessment allowed for the finding of a persistent visual acuity impairment at three years follow-up that is associated to abnormal foveal function and reduced retinal sensitivity, with resolution of morphological macular changes, thus suggesting that visual acuity changes can be associated to foveal outer layers of retinal neurodegeneration, similarly to that found by our group in several neurodegenerative diseases [5].

Bilateral visual impairment in a case of FSHD followed for three years was secondary to macular degeneration trait and not associated to vascular retinal barrier damage. The self-resolved macular abnormalities lead us to think that the pseudocyst changes could be an epiphenomenon of FSHD disease, possibly due to retinal vacuolization, whose pathogenic mechanisms need to be clarified by larger studies correlating signs of supposed vasculopathy and macular dystrophy appearance.

## Figures and Tables

**Figure 2 diagnostics-12-02977-f002:**
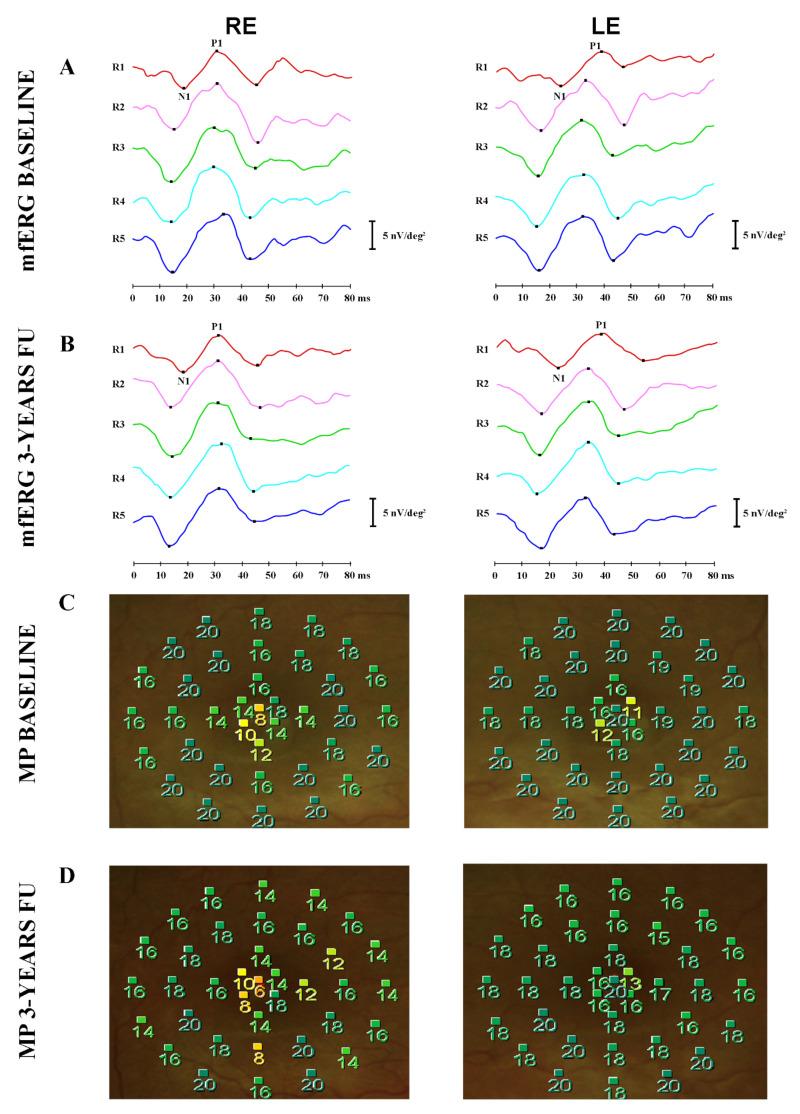
Macular function was investigated by multifocal electroretinogram (mfERG, VERIS Clinic TM 4.9 Electro-Diagnostic Imaging, San Mateo, CA, USA) and microperimetry (MP1, Nidek, Gamagori, Japan). The mfERG recordings, collecting signals derived from photoreceptors and bipolar cells, showed bilateral reduction of the response amplitude density (RAD) in Ring 1, between 0–2.5° degrees centered on the fovea (**A**); mfERG recordings (**B**) remained abnormal for impaired foveal function (R1). Similarly, the microperimetry showed a reduction of mean sensibility only in the foveal area in both eyes (mean sensitivity at baseline was 17.1 dB in RE and 18.7 in LE) (**C**); microperimetry mean sensitivity was reduced to 15.2 dB in RE and 17.4 in LE as well (**D**).

## Data Availability

We have excluded this statement because the study did not report any data.

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
