# Peer review of "Morpho-Functional Macular Assessment in a Case of Facioscapulohumeral Muscular Dystrophy: Photoreceptor Degeneration as Possible Cause for Reduced Visual Acuity over Three Years of Follow-Up"

_diagnostics, 2022, doi:10.3390/diagnostics12122977_

Round 1

Reviewer 1 Report

In this case, Mariacristina et al. describe a case of facioscapulohumeral muscular dystrophy with baseline and follow-up measures, multimodal imaging and functional investigations. This is a good case study. I only have a few comments to improve an already excellent manuscript:

1. The authors describe fever and muscle weekness. Did the patient get a systemic work-up, or any systemic treatment?

2. Did the slit-lamp examination reveal redness, inflammation in the anterior chamber, or vitritis? If not, please also outline these negative findings in the manuscript to avoid any confusion.

3. Do you have a fundus photography from the 3 year follow-up also? And did the tortuosity change from baseline to 3 year follow-up?

Reviewer 2 Report

This is such an interesting study. There's a considerable dearth of knowledge regarding the macular phenotypes of FSHD. This case study provides an essential primer for more in-depth studies on the macular phenotypes of FSHD. In addition, it sheds light on the plausible causes of reduced visual acuity in FSHD. 

  1. Please report the age at which FSHD had its onset. 
  2. Also, how was she diagnosed to have D4Z4 repeat contraction? 
  3. Please suggest if the patient has ptosis? 
  4. Page 6, lines 13–16: can there be any other possible root for the visual acuity impairment? 
  5. Previous studies suggest that patients with FSHD have thinner central corneas and lower intraocular pressure measurements. It would be beneficial for the readership if the authors could comment on this based on their findings. 
  6. I assume the image qualities were reduced during pdf conversion. The images in the pdf document this reviewer found were not of the best quality. 

Round 2

Reviewer 2 Report

Please mention the method used for diagnosis.

Author Response

Molecular diagnosis was made by analyzing D4Z4 repeat length on chromosome 4q35 by DNA testing from peripheral blood.

All these information were derived from additional patient’s reports requested by the reviewer and added in the revised manuscript.